# Pharmacological Enhancement of Regeneration-Dependent Regulatory T Cell Recruitment in Zebrafish

**DOI:** 10.3390/ijms20205189

**Published:** 2019-10-19

**Authors:** Stephanie F. Zwi, Clarisse Choron, Dawei Zheng, David Nguyen, Yuxi Zhang, Camilla Roshal, Kazu Kikuchi, Daniel Hesselson

**Affiliations:** 1Diabetes and Metabolism Division, Garvan Institute of Medical Research, Darlinghurst, NSW 2010, Australia; stephzwi@gmail.com (S.F.Z.); clarisse.choron@estbb.ucly.fr (C.C.); d.nguyen@garvan.org.au (D.N.); jialhs@hotmail.com (Y.Z.); camillaroshal@hotmail.com (C.R.); 2Developmental and Stem Cell Biology Division, Victor Chang Cardiac Research Institute, Darlinghurst, NSW 2010, Australia; d.zheng@victorchang.edu.au; 3St Vincent’s Clinical School, University of New South Wales, Kensington, NSW 2052, Australia

**Keywords:** regulatory T cell, zebrafish, small molecule screen, pramipexole, dopamine signaling

## Abstract

Regenerative capacity varies greatly between species. Mammals are limited in their ability to regenerate damaged cells, tissues and organs compared to organisms with robust regenerative responses, such as zebrafish. The regeneration of zebrafish tissues including the heart, spinal cord and retina requires *foxp3a+* zebrafish regulatory T cells (zTregs). However, it remains unclear whether the muted regenerative responses in mammals are due to impaired recruitment and/or function of homologous mammalian regulatory T cell (Treg) populations. Here, we explore the possibility of enhancing zTreg recruitment with pharmacological interventions using the well-characterized zebrafish tail amputation model to establish a high-throughput screening platform. Injury-infiltrating zTregs were transgenically labelled to enable rapid quantification in live animals. We screened the NIH Clinical Collection (727 small molecules) for modulators of zTreg recruitment to the regenerating tissue at three days post-injury. We discovered that the dopamine agonist pramipexole, a drug currently approved for treating Parkinson’s Disease, specifically enhanced zTreg recruitment after injury. The dopamine antagonist SCH-23390 blocked pramipexole activity, suggesting that peripheral dopaminergic signaling may regulate zTreg recruitment. Similar pharmacological approaches for enhancing mammalian Treg recruitment may be an important step in developing novel strategies for tissue regeneration in humans.

## 1. Introduction

The capacity for regeneration varies greatly between species [1]. While humans are limited in their ability to replace lost cells, tissues and organs, regeneration occurs naturally in many fish and amphibian organs [2]. It is suggested that in mammals, an evolutionary trade-off occurred whereby regenerative ability was lost in many tissues with the emergence of more potent and complex immune responses [3]. In contrast, zebrafish have retained the ability to regenerate complex organs in a regulatory T cell (Treg)-dependent manner [4]. Interestingly, Treg-dependent mechanisms are increasingly implicated in the limited repair and regeneration processes observed in mammals [5]. Thus, understanding mechanisms of Treg-mediated tissue repair in a highly regenerative species, such as zebrafish, may provide the foundation for novel approaches toward expanding the scope of tissue regeneration in humans.

Regeneration is the process whereby damaged cells or body parts are replaced. Adult mammals constitutively undergo homeostatic regeneration in certain tissues (e.g., hair and skin) [3]. However, humans are strikingly limited in their capacity for injury-induced or facultative regeneration, the process of tissue replacement following significant trauma, such as amputation [6]. Definitive explanations for the vast regenerative differences between organisms remain elusive. One possibility is that certain genetic modules are present and functional in highly regenerative species but not in poorly regenerative ones [7]. Alternatively, epigenetic mechanisms regulating injury-activated gene expression may explain differential regenerative capacity [8,9].

Interestingly, injury-induced regeneration is present during mammalian development but largely disappears as animals approach developmental maturity [10]. For example, foetal and postnatal mice can regenerate cardiac muscles, while adult mice display little or no cardiac regeneration [11,12]. Similarly, children demonstrate the ability to regenerate lost fingertips, a process lost to adult humans [13,14]. These data suggest that reactivating developmental programs in non-regenerative species, such as adult humans, may require fewer manipulations than previously expected [15]. Thus, appropriate animal models, particularly those that share key facets of vertebrate biology, may identify possible regenerative approaches for humans. The zebrafish is one such model as the genome shows 70% homology to humans [16]. In addition, rapid development, transparent and chemically permeable skin, and ease of genetic tagging with fluorescent markers offer significant advantages for in vivo small molecule screening [17].

Tregs are centrally implicated in vertebrate tissue repair [18]. In humans, they are necessary for a fully functional adaptive immune system and require the transcription factor Forkhead Box P3 (FOXP3) to functionally differentiate [19]. In mice, Tregs accumulate rapidly in injured skeletal muscle and infected lung tissue, where they promote the release of growth factors necessary for cell differentiation and tissue repair [18,20]. In zebrafish, the FOXP3 homolog *foxp3a* is among the most upregulated regeneration genes, and Tregs accumulate in the damaged spinal cord, heart and retina, where they release essential proregenerative factors to stimulate precursor cell proliferation [4]. Zebrafish also express *foxp3b*, another FOXP3 ortholog, at low levels although it is not detected in mature T cells and its function remains unclear [21]. Zebrafish Tregs (zTregs) demonstrate remarkable tissue-specificity by releasing factors tailored to the precursor cell-type present in the damaged tissue. Furthermore, zTreg ablation severely impairs regenerative outcomes, functionally demonstrating that zTregs are required for zebrafish organ regeneration [4].

Increasing Treg numbers in damaged tissues by enhancing recruitment and/or promoting their expansion in situ may be an important step(s) in unlocking the potential for regenerative tissue repair in humans. At present, there are no pharmacological agents that stimulate zTreg recruitment. The current study seeks to identify modulators of zTreg infiltration to injury sites via a high throughput screen of the NIH Clinical Collection, a library of small molecules with a history of previous use in clinical trials. To this end, we optimized the zebrafish tail amputation assay to visualize transgenically labelled zTregs in actively regenerating tissue. Identification of molecules capable of enhancing zTreg recruitment may be useful in defining the next steps for activating latent regenerative potential in humans.

## 2. Results

### 2.1. Chemical Screen for Modulators of zTreg Recruitment to Regenerating Tail Tissue

We established an in vivo screening platform for the quantification of zTregs that were recruited to regenerating fin tissue by three days post-injury (dpi) (Figure 1a–c). We used this model to perform an in vivo chemical screen to identify agents capable of modulating zTreg recruitment after injury. We administered the NIH Clinical Collection of small molecules with a history of use in clinical trials to 4 weeks post-fertilization (wpf) juvenile zebrafish at 10 μM for 3 dpi. The number of *foxp3a*+ cells in the regenerative site was quantified (Figure 1d). Drugs with Z-scores above 1.96 (10 of 727 compounds) were retested at 5 μM and 10 μM using the same conditions as the initial screen. Only NCP002453_A11 significantly increased zTreg recruitment in the retest (Figure 1e). In addition, we identified 44 molecules that potently inhibited zTreg recruitment in the primary screen (Table A1). Importantly, 14/44 zTreg recruitment inhibitors were known immune suppressants (Table A1), validating that the screen was capable of detecting chemical modulators of T cell function.

### 2.2. Pramipexole Stimulates zTreg Recruitment in an Injury-Dependent Manner

To validate the results of the screen, NCP002453_A11 (pramipexole) was ordered from an independent supplier and tested in the amputation model (Figure 2a). Pramipexole showed a maximal response at 5 μM, and the reduced activity at higher doses suggests that the therapeutic window for zTreg recruitment is between 1–10 μM (Figure 2b). To test whether pramipexole required signals from the injured tissue to induce zTreg accumulation, we treated uninjured zebrafish with 5 μM pramipexole and did not observe an increase in the number of *foxp3a+* zTregs in the tail after 3 days of treatment (Figure 2c).

### 2.3. Pramipexole Stimulates zTreg Recruitment via Dopaminergic Pathways

Pramipexole is a dopamine agonist that activates D_2_, D_3_ and D_4_ receptors with low nanomolar affinities [22]. We next tested whether activating a single dopamine receptor is sufficient to stimulate zTreg recruitment using apomorphine, a structurally distinct dopamine D_2_-selective receptor agonist [23]. However, apomorphine did not increase zTreg recruitment at the tolerated doses (Figure 3a), suggesting that zTreg recruitment may require activation of a specific dopamine receptor(s). To further explore the pathway by which pramipexole regulates zTreg recruitment, we co-treated pramipexole with the dopamine receptor antagonists SCH-23390 or amisulpride (with selective binding profiles for D1-like and D2-like receptors respectively). While SCH-23390 treatment alone did not impact zTreg recruitment, co-treatment blocked the effect of pramipexole (Figure 3b). Together these data suggest that pramipexole acts through diverse dopamine receptors to enhance zTreg recruitment.

## 3. Discussion

In this study, we established an in vivo screening platform to identify novel modulators of zTreg recruitment to injured and regenerating tissue. While many (~6%) compounds interfered with zTreg recruitment, only pramipexole increased the number of injury-infiltrating zTregs. We conclude that pramipexole acts primarily by enhancing zTreg recruitment as we did not observe local expansion in uninjured pramipexole-treated animals.

Pramipexole is an FDA-approved dopamine agonist [24] that is also reported to reduce oxidative stress and prevent apoptosis in Parkinson’s Disease (PD) models [25,26], although the precise neuroprotective mechanism remains unclear. Dopamine receptors are a family of G-protein coupled receptors that are grouped into two major subfamilies: D1-like receptors (D_1_ and D_5_) and D2-like receptors (D_2–4_) [27]. Both subtypes are highly conserved in vertebrates including zebrafish [28,29]. While dopamine receptors are predominantly expressed in the central nervous system, human Tregs also express dopamine D1-like and D2-like receptors [30]. The presence of dopaminergic receptors on Tregs suggests that pramipexole may have direct effects on zTregs in our recruitment assay, providing additional insight into unanticipated and possibly beneficial additional anti-PD activities of this dopamine agonist.

We attempted to identify whether a specific dopaminergic receptor is required for pramipexole action. Given that pramipexole binds to D2-like receptors (D_2–4_) in humans [24], we first attempted to mimic the effect of pramipexole by administering the D_2_-specific agonist apomorphine. However, D_2_ agonism with apomorphine was not sufficient to phenocopy the effects of pramipexole. In complementary experiments we blocked the effects of pramipexole via co-treatment with the D1-like class-selective dopamine receptor antagonist SCH-23390, which also has inhibitory effects on human Tregs [30]. This result was unexpected since pramipexole is reported to show minimal activity towards D1-like receptors. One limitation of this analysis is that the receptor specificities of pramipexole and SCH-23390 may differ between humans and zebrafish. Nonetheless, our data suggest that multiple dopaminergic receptors mediate pramipexole action in zTregs. Future studies of genetically engineered models lacking individual receptors might help clarify the pramipexole mechanism of action.

Importantly, we showed that zTreg recruitment can be modulated by pharmaceutical agents. In addition to pramipexole, we also identified 44 drugs that had inhibitory effects. The largest class of inhibitors were broad immune suppressants, a finding that validates our screen. Interestingly, the second largest class of inhibitors were CNS agents, including two D_2_ receptor antagonists, thioridazine hydrochloride and chlorpromazine hydrochloride, as well as amoxapine, a tricyclic antidepressant whose metabolite 7-hydroxyamoxipine is a potent dopaminergic antagonist. These data further implicate dopamine signaling in zTreg biology.

In this study, we used the zebrafish tail regeneration paradigm due to its experimental compatibility with small molecule screening. Future studies should investigate whether pramipexole enhances zTreg recruitment to more complex tissues, such as the heart or spinal cord. Eventually, pharmacological enhancement of Treg-mediated tissue regeneration may be one path to enhancing the regenerative potential of poorly regenerating human organs.

## 4. Materials and Methods

### 4.1. Zebrafish Maintenance and Breeding

Zebrafish were maintained at the Garvan Institute of Medical Research and Victor Chang Cardiac Research Institute in Sydney, Australia. All procedures were approved by the Garvan Institute of Medical Research/St Vincent’s Hospital Animal Ethics Committee under Animal Research Authorities 15_13 (4 May 2015), 17_21 (7 August 2017) and 18_14 (18 July 2018). Larval zebrafish were housed in 1 L tanks (Tecniplast, Chester, PA, USA) and adult zebrafish in 3.5 L tanks (Tecniplast) with a maximum of 30 fish per tank. Fish were grown in 28 ± 1 °C recirculating chlorine-free water and were exposed to a day-night cycle of 14.5:9.5 hours light: dark and fed three times daily. Experimental animals were generated from weekly group matings. Fertilized embryos were incubated at 28.5 °C (Memmert, Büchenbach, Germany) in Embryo Media (0.03% (w/v) ocean salt (Aquasonic, Wauchope, Australia), 0.0075% (w/v) calcium sulphate (Sigma, St. Louis, MO, USA), 0.00002% (w/v) methylene blue (Thermo Fisher Scientific, Waltham, MA, USA) in water) at a density of 60 embryos per 25 mL dish (Thermo Fisher Scientific). At 3 days post-fertilization (dpf), zebrafish were screened for transgene expression using fluorescence microscopy (Leica DFC450 C), and at 1 wpf, larvae were placed into 1 L housing tanks in chlorine-free water, with the flow turned at 2 wpf.

### 4.2. Zebrafish Transgenic Lines

Transgenic *TgBAC (foxp3a: TagRFP; cryaa:EGFP)^vcc3^* (*foxp3a*:RFP) (Hui et al., 2017) zebrafish were used to visualize zTregs in vivo by epifluorescence.

### 4.3. Juvenile Fin Amputation

Petri dishes (90 mm; Thermo Fisher Scientific) and 12-well plates (polystyrene flat bottom; Sigma) were coated with 1.5% low-melt agarose (Sigma) in E3 solution (1.72% (w/v) sodium chloride, 0.076% (w/v) potassium chloride, 0.29% (w/v) calcium chloride, 0.49% (w/v) magnesium sulfate in water), and stored at 4 °C until use to prevent injured tail tissue from sticking to culture plates after amputation. Four wpf zebrafish were starved overnight to empty the digestive system and then anaesthetized with 0.4% Tricaine (ethyl 3-aminobenzoate methanesulfonate (Sigma), 20 mM Tris-HCl, pH 7.0) in Petri dishes, and a carbon steel scalpel blade (Swann–Morton, Sheffield, UK) was rolled over the tail region. Amputated zebrafish were rinsed with Embryo Media lacking methylene blue and dispensed into agarose-coated 12-well plates.

### 4.4. Chemical Screen and Drug Treatments

Amputated zebrafish (4 wpf) were treated with the NIH Clinical Collection (Evotec, Hamburg, Germany: 1 mM in DMSO) at a 1:100 final dilution of Embryo Media in 12-well plates. Treated fish were incubated at 28.5 °C and *foxp3a*+ cells in the regenerating region were counted under epifluorescence at 3 dpi. Each drug was screened in duplicate on different screen weeks. Retests and dose–response determination was performed under the same conditions as the screen.

### 4.5. zTreg Quantification

At 3 dpi zTregs (*foxp3a*: RFP+) in the regenerating tail region were visualized by epifluorescence (Leica DFC450 C). Cells were counted within the region spanning the amputation site to the tail tip. The *foxp3a*: RFP+ cells were directly counted in live animals.

### 4.6. Statistical Analysis

Statistical analysis was performed using GraphPad Prism 7 (San Diego, CA, USA). Cell counts were normalised to DMSO-treated controls for each plate and Z-scores were generated for each drug to select primary hits for retesting. One-way ANOVA with multiple comparisons was used to compare means between treatment and control groups. Multiple comparison-corrected *p*-values of 0.05 or less were considered significant.

## Figures and Tables

**Figure 1 ijms-20-05189-f001:**
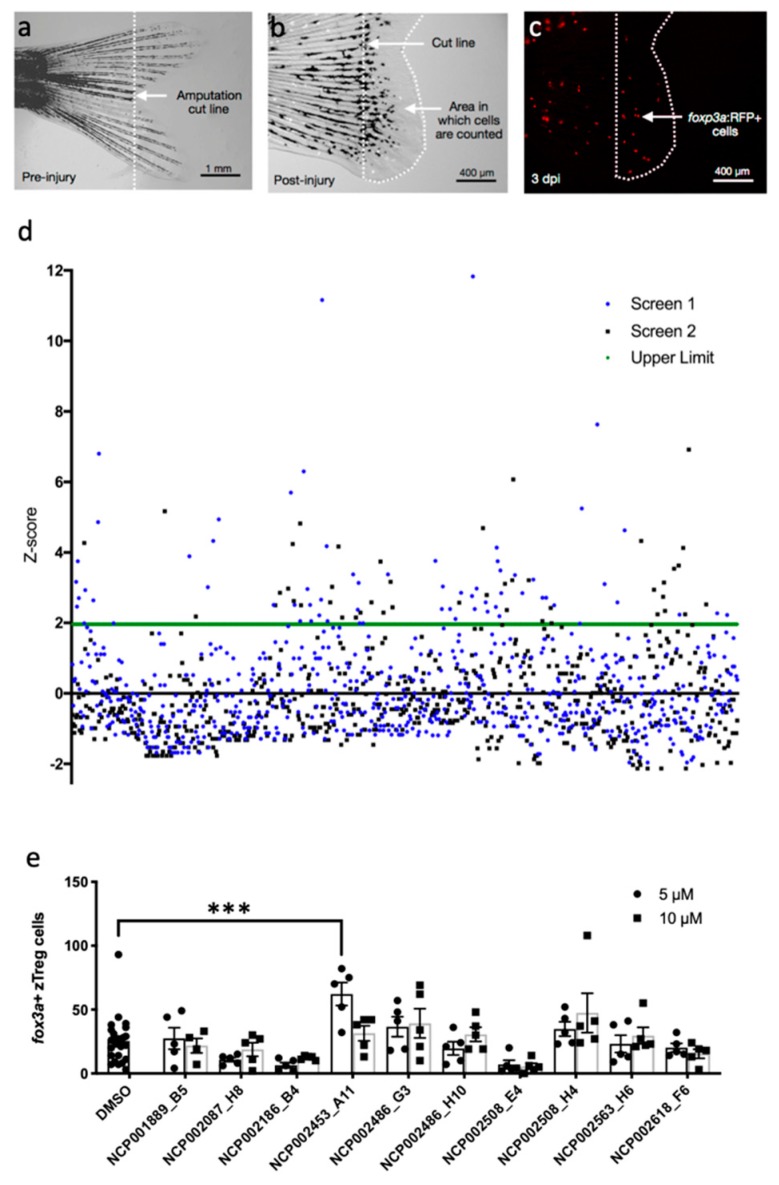
Chemical screen for modulators of zebrafish regulatory T cell (zTreg) recruitment in the regenerating tail. (**a**) Zebrafish tails were amputated at 4 weeks post-fertilization (wpf) along the dotted cut line. (**b**) Quantification was performed on the region from the amputation site to the tip of the regenerated tail at 3 days post-injury (dpi). (**c**) *foxp3a*: RFP+ zTreg cells were visualized by epifluorescence. (**d**) Normalized numbers of zTreg cells for each compound in the NIH Clinical Collection. Green line, statistical threshold for selecting compounds for retesting. (**e**) Screen hits were retested at 5 μM and 10 μM. Data presented as mean ± SEM; *n* = 5 per treatment group, *n* = 25 for controls. *** *p* < 0.001. None of the other treatments significantly increased the number of zTreg cells.

**Figure 2 ijms-20-05189-f002:**
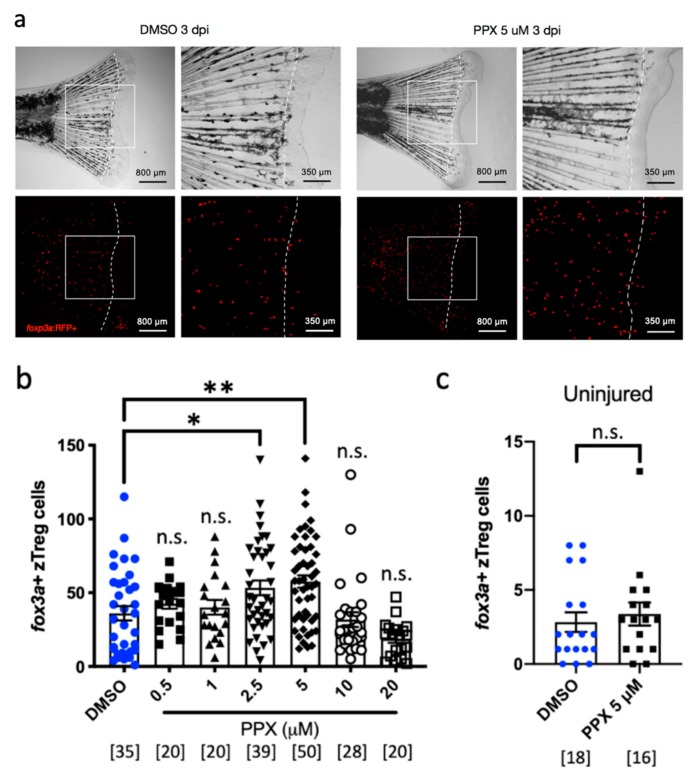
Pramipexole (PPX) enhances zebrafish regulatory T cell (zTreg) recruitment in regenerating tail tissue. (**a**) Regenerating tissue from DMSO- and PPX-treated zebrafish at 3 days post-injury (dpi). White dotted lines represent the amputation line. Cells distal to this line were counted. (**b**) Dose–response for PPX (0.5–20 μM) in the tail amputation assay. Quantification of *foxp3a*+ cells (mean ± SEM). (**c**) Quantification of *foxp3a*+ cells (mean ± SEM) in uninjured tail tissue from zebrafish treated with PPX or DMSO for 3 days. n.s., non-significant; * *p* < 0.05, ** *p* < 0.001. Number of animals in each group indicated in [].

**Figure 3 ijms-20-05189-f003:**
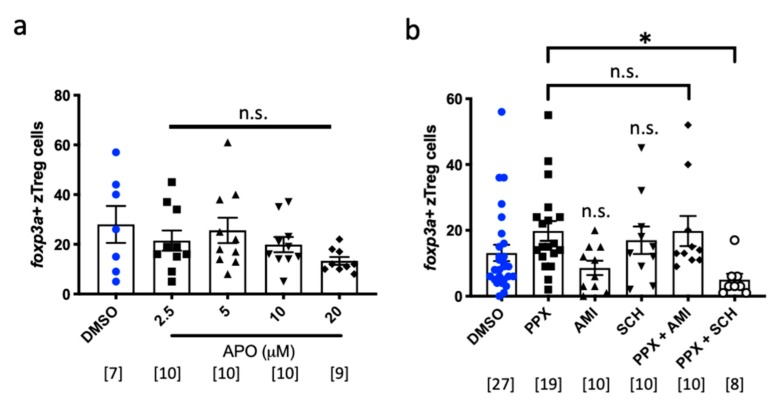
Role of dopamine signaling in zebrafish regulatory T cell (zTreg) recruitment. (**a**) Dose–response for apomorphine (APO) (2.5–20 μM) and (**b**) (co)treatments with dopamine antagonists in the tail amputation assay. Quantification of *foxp3a*+ cells (mean ± SEM). PPX, pramipexole; AMI, amisulpride; SCH, SCH-23390; n.s., non-significant; * *p* < 0.05.

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
