# Peer review of "Pharmacological Enhancement of Regeneration-Dependent Regulatory T Cell Recruitment in Zebrafish"

_ijms, 2019, doi:10.3390/ijms20205189_

Round 1

Reviewer 1 Report

This manuscript entitled "Pharmacological enhancement of regeneration dependent regulatory T cell recruitment in zebrafish" is interest and well-designed. However, there are some comments to be revised or added for publication. The authors have to carefully read those comments and can revise for improved manuscript.

1. The resulting finds may be simply speculation, and the real mechanism is unclear. Treg cells are known to be promoting regeneration by producing tissue-specific growth factors. So, I like to recommend determining the tissue-specific growth factors.

2. What was the rationale of choosing the indicated concentration in Fig. 1e? Also, treatment with 10 and 20 μM of PPX why did not affect zTreg recruitment?

3. Previously published study indicated that zTreg cells in injured zebrafish showed the highest expression at 7 dpi. In this study, why did the author choose at 3 dpi to determine zTreg quantification?

4. In Fig. 2, the image of fluorescence microscopy is of poor quality and must be improved. In addition, result of fluorescence microscopy should be included in Fig. 3.

5. Discussion section is poorly written. Limits and strengths of this study should be better mentioned in the discussion section. Also, the method section should be described as more detailly.

6. All the chemicals and reagents used in this study should be mentioned along with the details of manufacturers/suppliers.

Author Response

Thank you for the constructive comments. Please find our point-by response below.

The resulting finds may be simply speculation, and the real mechanism is unclear. Treg cells are known to be promoting regeneration by producing tissue-specific growth factors. So, I like to recommend determining the tissue-specific growth factors.

We agree that identification of fin-specific growth factors is an interesting line of research. However, that is a multi-year endeavour that is beyond the scope of this manuscript which focuses on the development of a new model to identify small molecules with proregenerative potential.

What was the rationale of choosing the indicated concentration in Fig. 1e? Also, treatment with 10 and 20 μM of PPX why did not affect zTreg recruitment?

We performed the primary screen at 10 μM. Libraries stored in DMSO can exhibit some loss of potency over time with repeated freeze/thaw cycles. Therefore we also performed the retests with 5 μM of pristine compound to compensate for this effect. In our experience every small molecule has a therapeutic window above which it is ineffective or even shows opposing effects due to toxicity (unless it has very poor solubility). We added a comment on this effect to the results.

Previously published study indicated that zTreg cells in injured zebrafish showed the highest expression at 7 dpi. In this study, why did the author choose at 3 dpi to determine zTreg quantification?

It is correct that we previously showed maximum zTreg recruitment to heart, spinal cord and retina at 7 dpi in adult zebrafish (Hui et al., 2017). However, in preparation for the small molecule screen we performed pilot experiments that indicated that 3 dpi is the optimal time point for analyzing zTreg recruitment to injured fin in juvenile (4 wpf) fish.

In Fig. 2, the image of fluorescence microscopy is of poor quality and must be improved. In addition, result of fluorescence microscopy should be included in Fig. 3.

The fluorescence images clearly show the individual zTregs that we quantified, particularly in the higher magnification panels. We think it is also important to show the lower magnification overview of the entire tissue. It is possible that the embedded images have reduced resolution so we will provide the original PPT files with the revision. The fluorescence images show representative data for one treatment condition (5 μM PPX and DMSO). We do not think that adding images for every dose and treatment would improve the clarity of the manuscript. As described in the Methods, quantification was performed under epifluorescence with live animals (not on images).

Discussion section is poorly written. Limits and strengths of this study should be better mentioned in the discussion section. Also, the method section should be described as more detailly.

We have modified the discussion in response to comments from the other reviewer. In addition we now discuss a limitation of using chemical probes in different species. We have added reagent information to the methods as requested below.

All the chemicals and reagents used in this study should be mentioned along with the details of manufacturers/suppliers.

Additional reagent information has been added to the Methods section.

Reviewer 2 Report

In this study, the Authors explore the possibility of enhancing zTreg recruitment with pharmacological interventions using the well-characterized zebrafish tail amputation model to establish a high-throughput screening platform. They discover one dopamine agonist, pramipexole, able to enhance zTreg recruitment after injury. The results of this work are of broad interest as Treg enhancement could be used in developing novel strategies for tissue regeneration in humans. The work is well-written; however, the rationale of the results is slightly difficult to follow. I found major limitations in the present form, especially in the validation of the D1 and D2 receptors involvement. Moreover, I would suggest to better detail the regeneration process following drugs administration and Treg recruitment (e.g. proliferation assays, neutrophils/macrophages recruitment). Thus, I strongly recommend to improve these parts before acceptance in IJMS.

Abstract

Line 20: the transgenic TgBAC(foxp3a:TagRFP; cryaa:EGFP)vcc3 (foxp3a:RFP) line has been generated by Hui et al., 2017 not by the authors. This is not clearly stated in the abstract.

Introduction

Line 19: the authors introduce the zebrafish gene fox3a that has a paralog fox3b. To be thorough, I would suggest to cite also the paralog.

Results

Line 38: The authors analysed the Treg activation at 3 dpa. However, the regeneration assays comprise different time points. I would suggest to perform a time course of Treg recruitment, considering also previous (and possibly later) stages post amputation.

Figure 2: The dose-response assay for PPX ranges from 0.5 to 20 μM. The increase in Treg recruitment is statistically significant at the doses of 2.5 and 5 μM but is decreased at higher doses. How the authors explain these opposite effects? In the graph statistically n.s. comparisons should be highlighted.

Line 12: the authors claim that: “apomorphine did not increase zTreg recruitment at the tolerated doses, suggesting that zTreg recruitment may have receptor sub-type specificity”. What are the basis of the assumption? I would suggest to better clarify this.

Line 15: since only the SCH-23390 dopamine D1-selective receptor antagonist blocks the effects of PPX on Treg recruitment and the use of a distinct dopamine D2-selective receptor agonist apomorphine does not enhance Treg recruitment, one could conclude that the effect of PPX is D1 specific. Is this the case? Is it possible to test a specific D1 agonist for Treg recruitment to prove this?

Moreover, the authors did not comment about the amisulpride effects and no statistic has been shown in the graphs of Figure 3a and partially missing in the figure 3b (while in the figure legend is reported n.s.).

Discussion

Line 20: The sentence: “The presence of dopaminergic receptors on Tregs suggests that pramipexole may have direct effects on zTregs in our recruitment assay, providing additional insight into its possible anti-PD activities” is misleading. The authors suggest that possible anti-PD activities have been exerted by Treg cells or by pramipexole?

Methods

General comments about methods: how many fish have been considered for each category/dose of drug? In addition, how long was the drug administration period and how the authors calculate the sufficient period of drug exposure? The authors have to insert this information in the methods.

Line 24: Why the fish might be starved before fin amputation?

I have no access to supplementary data.

Author Response

Thank you for the constructive comments. Please find our point-by response below.

Abstract

Line 20: the transgenic TgBAC(foxp3a:TagRFP; cryaa:EGFP)vcc3 (foxp3a:RFP) line has been generated by Hui et al., 2017 not by the authors. This is not clearly stated in the abstract.

The corresponding authors of the current manuscript were also the senior authors of Hui et al., 2017 paper describing the foxp3a:RFP transgenic line. The details of the zTreg transgenic line are provided in the methods section.

Introduction

Line 19: the authors introduce the zebrafish gene fox3a that has a paralog fox3b. To be thorough, I would suggest to cite also the paralog.

We added information on the foxp3b ortholog to the introduction.

Results

Line 38: The authors analysed the Treg activation at 3 dpa. However, the regeneration assays comprise different time points. I would suggest to perform a time course of Treg recruitment, considering also previous (and possibly later) stages post amputation.

We apologize for the confusion. All experiments were performed at the same time points with amputation occurring at 4 weeks post fertilization and analysis at 3 days post injury. We updated the text to consistently use 3 dpi to describe the analysis. We performed pilot experiments to determine that maximum recruitment occurred by 3 dpi.

Figure 2: The dose-response assay for PPX ranges from 0.5 to 20 μM. The increase in Treg recruitment is statistically significant at the doses of 2.5 and 5 μM but is decreased at higher doses. How the authors explain these opposite effects? In the graph statistically n.s. comparisons should be highlighted.

In our experience every small molecule has a therapeutic window above which it is ineffective or even shows opposing effects due to toxicity (unless it has very poor solubility). We added a comment on this effect to the results. We have also added n.s. to the relevant comparisons as requested.

Line 12: the authors claim that: “apomorphine did not increase zTreg recruitment at the tolerated doses, suggesting that zTreg recruitment may have receptor sub-type specificity”. What are the basis of the assumption? I would suggest to better clarify this.

Pramipexole activates D2,D3 and D4 receptors whereas apomorphine which only activates D2 is ineffective in this assay. Therefore we conclude that simply activating a 'dopamine receptor' is not sufficient and the specific receptor(s) may be important for understanding pramipexole activity. We clarified this point in the results.

Line 15: since only the SCH-23390 dopamine D1-selective receptor antagonist blocks the effects of PPX on Treg recruitment and the use of a distinct dopamine D2-selective receptor agonist apomorphine does not enhance Treg recruitment, one could conclude that the effect of PPX is D1 specific. Is this the case? Is it possible to test a specific D1 agonist for Treg recruitment to prove this?

This is an intriguing hypothesis that we now discuss in more detail in the discussion. It was an unexpected result since pramipexole is reported to be D2-specific and deserves further investigation in future studies.

Moreover, the authors did not comment about the amisulpride effects and no statistic has been shown in the graphs of Figure 3a and partially missing in the figure 3b (while in the figure legend is reported n.s.).

We now indicate the non-significant treatments in the graphs. We added a comment about amisulpride to the Results.

Discussion

Line 20: The sentence: “The presence of dopaminergic receptors on Tregs suggests that pramipexole may have direct effects on zTregs in our recruitment assay, providing additional insight into its possible anti-PD activities” is misleading. The authors suggest that possible anti-PD activities have been exerted by Treg cells or by pramipexole?

We clarified this sentence to indicate that effects on Tregs could be an unanticipated beneficial effect of pramipexole treatment in PD.

Methods

General comments about methods: how many fish have been considered for each category/dose of drug? In addition, how long was the drug administration period and how the authors calculate the sufficient period of drug exposure? The authors have to insert this information in the methods.

Please note we proved individual animal data points in all graphs and the number of fish in the Figure 1 legend. We now also indicate the total numbers in the graphs and/or figure legends for Figures 2 and 3. The drugs were administered immediately after amputation and the animals were exposed for three days until analysis. We expect that the dose/duration chosen may have generated some false negative results but was based on our previous experience designing in vivo small molecule screens.

Line 24: Why the fish might be starved before fin amputation?

We performed the drug treatments in 12-well plates. Fed fish would generate waste that would be toxic in these small chambers over 3 days. The purpose of overnight starvation is now noted in the Methods.

Round 2

Reviewer 1 Report

This revised manuscript is a well-written paper containing interesting results and scientific knowledge which merit publication. Also, this revised manuscript was correctly revision. Thus it is acceptable in the present version.

Reviewer 2 Report

The authors improved the descriptions in the text without adding additional experiments (e.g. more controls and different time points of analyses). These data, although not strictly necessary, would have enhanced the significance of the work. Moreover, the data about the specificity of the different dopamine receptors would have been experimentally improved without adding huge amount of work, money and time. This would have increased the interest of the work in terms of translational potential of the research. The manuscript in the present form is acceptable for the publication although, collectively, these issues dampen my enthusiasm for the relevance of these studies to the disease (Parkinson).